# Biomimicry-Gradient-Based Algorithm as Applied to Photonic Devices Design: Inverse Design of Flat Plasmonic Metalenses

**Kofi Edee** [1,2]

1 Institut Pascal, Université Clermont Auvergne, BP 10448, F-63000 Clermont-Ferrand, France; kofi.edee@uca.fr
2 CNRS, UMR 6602, Institut Pascal, F-63177 Aubière, France

**Abstract:** The classical adjoint-based topology optimization (TO) method, based on the use of a random continuous dielectric function as design variable distribution is known to be one of the timely efficient and fast optimization methods enable a very high performance functional optical devices. It relies on the computation of the gradient of a figure of merit (FOM) with respect to the design parameters. The gradient of the figure of merit (FOM) may then be used to update the design vector element in several senarios. One of the most common use scenario consists of updating simultaneously all the design parameter vector elements. In a linear problem case involving a simply convex FOM-function shape, using the gradient information, it is a relatively easy to reach an optimal solution. In the case of constrained and non linear problems stated in an infinite and indeterminate design space, the conventional TO, a local optimizer, may require multiple restarts, with multiple initial points and multiple runs. The algorithm strongly depends on the initial conditions. In this paper, we report a global-like optimizer inspired by a wolf pack hunting, enabling efficient design of metasurfaces through their geometrical parameters. We apply the method to design a non periodic metasurface consisting of plasmonic metalenses, enabling a high energy flow focusing on a well-defined 2D focus spot. Numerical results show that the proposed inverse design method has a low sensitivity to initial conditions. In our design method of metalens, we optimize the full micro device at once, and demonstrate that the proposed method may provide both symmetric and more creative unexpected asymmetric on-axis metalenses even though under a normal illumination.

**Keywords:** inverse design; metasurfaces; metalens

## 1. Introduction

In the nanophotonics field, wavefront design is one of the timely ways of enabling expected functionalities for a flat and thin metasurface. These flat functional metasurfaces provide many applications such as beam deflection or focusing, etc., in which high efficiencies devices are required. The inverse design method based on forward and adjoint computations [1–8], has proven to be one of the most efficient in practice for achieving high performance optimized structures. Like any gradient-based method, the adjoint-based-inverse-design method relies on the computation of the gradient of a objective function with respect to the design parameters. The gradient of the figure of merit (FOM) is used to modify the design parameters in order to increase the figure of merit (FOM). The key point, hence the main advantage of the adjoint-based strategy, is that it allows an evaluation of the gradient of the FOM with a cost independent of the number of the design parameters. In addition, this gradient vector is used, at each iteration, to simultaneously update all the design parameters. The simultaneous-pushing process of design parameters, in each iteration, lacks randomness and this may trap the gradient-based algorithm in basins of local minimum. In linear problems with a simply convex shape merit function, using the gradient information, it is relatively easy for some gradient-based algorithms to reach an optimal solution. Therefore these gradient-based algorithms are eminently suitable for finding an optimal solution when the starting point is close to a local minimum. However,

in photonics, one often faces constrained and non linear problems stated in an infinite and indeterminate design space. Therefore, the conventional gradient-based optimization method requires multiple restarts initial points and multiple runs, hoping that a best single solution may be discovered at the end of these runs. This process may require higher computing memory capacity. On the other hand, global optimizers, continue to emerge, providing researchers a large choice of tools, in several disciplines. Some of them, enable an efficient random sampling of the design space, in each iteration, in order to detect and always maintain several better solutions during the optimization. This is the case for some metaheuristic algorithms [9–15]. Referring to the research-strategy, optimization algorithms are mainly based on four approaches: complete, random, heuristic, and metaheuristic. In the complete strategy, all possible combinations are evaluated in order to target the best solution. This strategy allows finding optimal solutions but it is computationally costly, which makes it impractical for high dimension problems. The random search is a way to attempt to overcome the drawback raised by the complete research-strategy. This strategy considers many initial candidates as possible solutions. In the best case, an optimal solution can be found earlier among these candidates, in the optimization process. However, this random strategy may behave as a complete method in its worst case. In heuristic algorithms, heuristic information which highly depends on the problem at hand is enforced in order to indicate certain ways to find optimal solutions. This strategy cannot be generalized since each problem involves (requires) a specific heuristic information. In the metaheuristic algorithm, few assumptions about the optimization problem being solved are made. This makes it possible to apply them for a variety of problems contrary to the heuristic algorithms. These methods range from Genetic algorithm [9], particle swarm optimization [10], ant colony optimization [11], differential evolution (DE) [12], grey wolves, whales and Harris Hawks [13–15], and the slime mould algorithm [16]. In general, they can be performed based on two approaches: a single-solution-based approach and a population-based approach. In the single-solution approach, only a single candidate (one solution) is generated, modified and updated during the optimization process, while in the population-based approach, as its name indicates, a set of random initial candidates (population) are generated and evolved during each iteration of the optimization phase. This latter approach is mostly inspired by natural phenomena and may be usable for a variety of problems without the need for the gradient information of the objective function.

In this paper, we propose a new nature-inspired optimization method taking advantage of the information of the variation of the FOM, to evolve a single-initial generated possible solution at each iteration. The proposed method is then a semi-complete single-solution approach. The initial solution, the vector of the design parameters, is a vector of geometric parameters characterizing the structure to design. As a semi-complete approach, the proposed method focuses on modifying, at each iteration, some elements of this single candidate (vector), in order to improve the FOM. To reduce the computational time involved by this semi-complete strategy, we introduce a cooperative strategy inspired by one of the smartest predators, wolves, hunting in a pack, an escaping prey. The proposed wolf hunting-based algorithm is different from the grey wolf optimizer (GWO) [13] which is a population-based gradient-free optimization technique. In the proposed method, at the beginning of each iteration, the adjoint-based method (based on the reciprocity theorem) is used to compute the variation of the FOM with respect to the variation of the design parameters. This FOM variation is then associated with the fitness of the design parameters. In the topology optimization method based on homogenisation [17,18] or shape optimization [19], the gradient of the FOM, is used to update simultaneously the design parameters vector. This strategy consisting of simultaneously pushing the design vector elements at each iteration, enforces the local nature of this kind of optimizer. Contrary to local optimizers, in order to avoid the trap of local minima, in the proposed algorithm, two unmissable phases of a global optimizer, namely exploration phase and exploitation phase, are integrated. Once the sequence of fitness is evaluated, it is sorted from the most sensitive agent to the lowest and categorized into three chasing-teams, operating, at each

iteration, in exploitation phase or exploration phase. The exploration phase (soft research) is related to global search as well as exploitation (intensive research) is related to local deeply search.

In Section 2, we present the mathematical formulation of the proposed biomimicry concept. In Section 3, we demonstrate its ability to design plasmonic metalenses. In photonics, these devices generally refer to metallo-dielectric structures enabling focusing light based on surface plasmons polaritons (SPPs) excitation. Here, only one-dimensional micron-scale ($50\lambda$-width) flat devices are discussed and designed. The behaviour of these lenses are based on the extraordinary optical transmission (EOT) phenomenon [20]. This phenomenon consists of an enhancement of the transmission of light through a subwavelength perforated opaque metallic film. Any incident TM-polarized plane wave that hits the plasmonics film from free space, will be coupled into an upper and downer SPPs mode (thick film) or a IMI mode (tiny film), and that field will be heading towards the Fabry-Perrot cavity. While emerging through the slits array, the phase of that diffracted (transmitted) field can be completely manipulated by fine-tuning the slit widths or slit depths, resulting in a tailored lens with high resolution. Finally, in the case of symmetric optimized device, to provide ultimate savings in time, we introduce a principle, called in this paper a two-stitched-meta-cells principle. In this design strategy. the whole structure is treated as a concatenation of two sub-meta-cells, geometries of which can be deduced each from other by a simple symmetry. Only one sub-structure is specifically designed, as a non-periodic off-axis-meta-lens. In addition, the completed on-axis-metalens is obtained by stitching together the two meta-cells.

## 2. Methods: Concept and Mathematical Model

A wolf pack hunt can be viewed as a multi-agents system in which each wolf chases a prey accounting for the fitness of each agent participating in the hunt. Wolves are opportunistic predators which base their hunting process on a collective strategy. Wolves may be considered to be operating in a pack when there are more than one wolf looking for prey. The main rule controlling the movement of each wolf in the whole pack is not to rely on a predefined hierarchy in the group but, from this collective behavior, emerges a spontaneous hierarchy based on the fitness of each agent to achieve the task properly by accounting for the locations of the other members, with respect to the prey's position. This assumes that, at each phase of the hunting process, each agent, while being autonomous, is able to access and process main information concerning its own position and also the positions of the other agents, with respect to the prey's location. This hunting strategy can be roughly divided in three phases:

1. Location phase
2. Stalking phase
3. Chasing phase

### 2.1. Location Phase

In this initial phase, the pack of wolves try to locate a prey or a pack of preys. The efficiency of the wolf pack to locate prey in a given geographic location strongly depends on several factors including the size of the wolves pack (number of wolves in the pack) and the size of the territory occupied by the pack. In the initialization phase of the hunting, let us consider a pack of $N_p + 1$ randomly distributed wolves occupying $d$-width territory. Let us consider there is a reference frame in which each wolf's location $(x_k)_k$ in the pack may be defined with respect to an origin point $x_0$. Here, we consider a one-dimensional problem and we denote $e_k$ the distance between the agent labelled $k$ and its closest neighbor $k - 1$. The geographic distribution of the wolf pack can be characterized by a sequence

of $N_p$-tuple of random variables $[e_k]_k$ inside a range $[e_{min}, e_{max}]$. These random variables must satisfy $\sum_{k=1}^{N_p} e_k = d$ and they can be generated through the following formula:

$$\begin{cases} e_k = (e_{max} - e_{min})r_k + e_{min}, \text{ generation} \\ e_k = d\dfrac{e_k}{\sum_{q=1}^{N_p} e_q}, \text{ normalization} \end{cases} \quad (1)$$

where $(r_k)_k$ is a sequence of random variables belonging to $[0,1]^{N_p}$. As displayed in Figure 1, the sequence of the position $(x_k)_k$-variables is linked to $[(e_k)_k]_k$ as:

$$x_k = x_{k-1} + e_k, k \in [1, N_p], x_0 = 0, x_{N_p} = d. \quad (2)$$

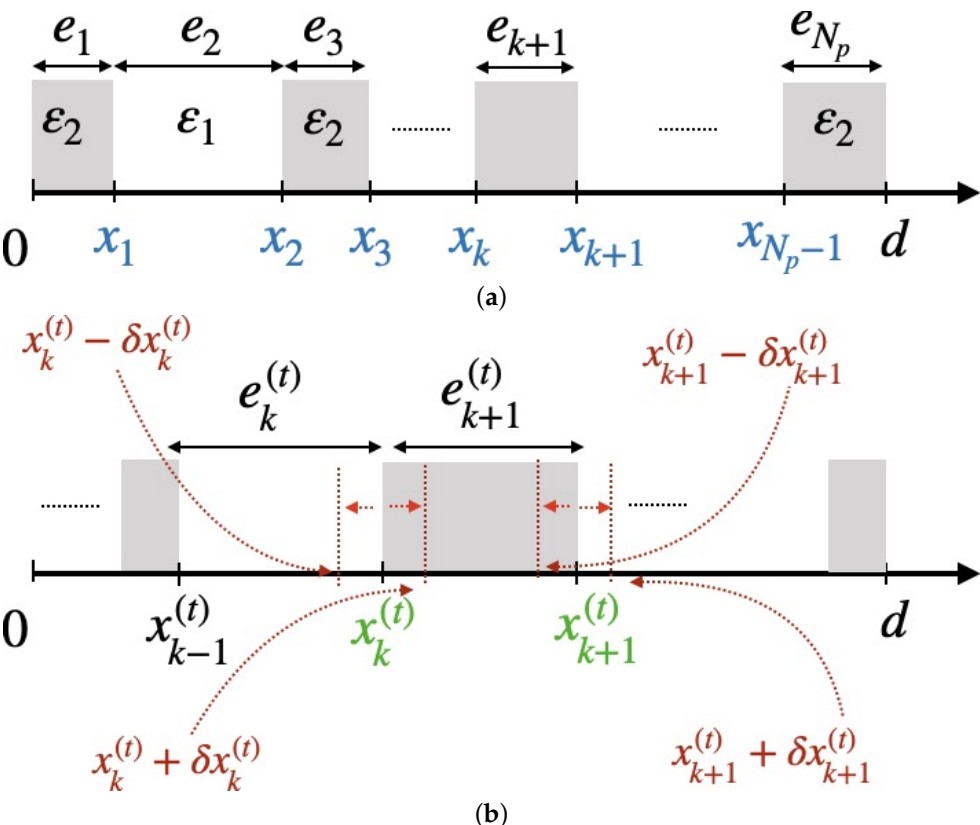

**Figure 1.** (**a**) Sketch of variables $e_k$ distribution. (**b**) Sketch of design variables $x_k$. In the proposed optimization process, the increment of $x_k$ increasing favorably the FOM is kept as the best current optimal profile.

Initially, the whole pack operates in exploration mode until the prey is located. Once the prey is located, the stalking phase starts and some parts of the hunters will transit from exploration to exploitation mode. The objective of the predator's pack is to capture at least one individual from the herd and this objective may be handled by a function called the figure of merit (FOM). At each step of the hunting process, the FOM depends on the sequence of the wolves positions $(x_k)_k$. The vector **X** which coordinates are $x_k$ is our single design variable vector, coordinates of which will be modified and updated during the optimization process.

### 2.2. Stalking Phase

The wolf pack begins the stalking phase by attacking the herd and trying to single out and tracking the most vulnerable individual of the herd. While carrying out the pursuit, each hunter can evaluate simultaneously its distance and the distance of the other

agents with respect to the prey's location. However, faced with the hunter's attack, the prey will try to escape and the prey's position will continuously change according to the predators actions. Let us denote $(x_k^{old})_k$ the sequence of each wolf position before the prey reaction and $(x_k^{new})_k$ the new sequence of $(x_k)_k$-variables when the prey operates some countermeasures. Let us recall that during the stalking phase, each wolf can estimate the fitness of any agent to help the group to achieve this collective task properly. Let us denote $(g_k)_k$ the sequence of fitness of each agent. Mathematically speaking, the vector $[g_k]$ contains the values of the variation of the FOM with respect to the variables $(x_k)_k$. The adjoint-based method shows their efficiency and effectiveness in computing the fitness parameters with a cost of computation time that does not depend on the number of the variables $x_k$. At the $t^{th}$ iteration, this variation of the objective function, denoted $g^{(t)(x_i)}$, at all nodes $x_i$ of the design area is computed with two simulations. The key point of the computation of $g^{(t)}$, already discussed in [21], is to consider that fictitious currents are induced when the system under consideration transits from a state called *old* to a state called *new*. This transition may be due to an evolution of the geometrical and/or the physical parameters of the system. A more detailed discussion on the computation of the gradient of the variation of the FOM, applied here, is provided in references [21]. Once the sequence of fitness is evaluated, it is sorted from most sensitive agent to the least and categorized into three chasing-teams as follows:

$$
\begin{cases}
\tilde{G}_1 = \{g_k \, / \, g_k \geq s_1\} \\
\tilde{G}_2 = \{g_k \, / \, s_2 \leq g_k \leq s_1\} \\
\tilde{G}_3 = \{g_k \, / \, g_k \leq s_2\}
\end{cases}
\tag{3}
$$

In the next phase, the chasing phase, a part of the pack constituted of wolves belonging to the $\tilde{G}_1$ and $\tilde{G}_2$ teams switches to an exploitation (intensive search) mode while agents of $\tilde{G}_3$-team remain in an exploration mode. In this paper, $s_1$ and $s_2$ are set to 25% and 1%, respectively.

*2.3. Chasing Phase*

If the selected prey escapes, the wolf pack will pursue it with the aim to capture it. Wolves are known to be very fast sprinters over very long distances. However, instead of relying on their athletic talents, wolves are able to craft a highly effective collective strategy by gradually encircling the prey until it stops moving. This part of the hunting process is driven by the two $\tilde{G}_1$ and $\tilde{G}_2$ teams using two different strategies. Wolves of the $\tilde{G}_1$-team constituted of hunting agents whose positions are the closest to the prey's location, attack and pursue the prey by pushing it towards the $\tilde{G}_2$-team's agents. The $\tilde{G}_2$ group are also close to the prey but slightly further compared to the positions of wolves operating in the $\tilde{G}_1$ team. In their exploitation mode, the wolves of the $\tilde{G}_1$ and $\tilde{G}_2$ teams push the prey towards the $\tilde{G}_3$ team, constituted of some agents pack lagging behind, which will capture the prey as a last resort, in the event this prey had the bad idea to turn back or in the case it turns round in circle. It is worth noting that here, the hierarchy of wolves that takes place during the hunting process does not follow the predefined natural hierarchy rules. During the hunt, wolves operate in the pack autonomously and can only be distinguished from each other by the hunting team they belong to. Moreover, they are interchangeable and the only information required for their classification and categorization into a given hunting team is the sequence of the sensitivity values $(g_k)_k$. This sequence is re-evaluated for the whole agents of the pack, at each time (iteration) $t$, as soon as the prey decides and performs an action. Therefore as the hunt evolves, any agent located at a position $x_k$ can switch from a group to another one, at any moment $t$, depending on the value of it sensitivity $g_k$ at this moment.

2.3.1. Team $\tilde{G}_1$ Strategy: Exploitation Mode with a Hard Besiege

Wolves in this situation are those with the highest sensibility, i.e., a fitness factor $g_k$ greater than a fixed threshold $s_1$. Each of them, can visualize and test the prey. They can

detect any weakness or vulnerability and thwart any of the prey's attempt to escape, by developing countermeasures, involving some rapid pounces if needed, or come back to its previous position. This behaviour can be formulated as follows:

$$x_k^{update} = \begin{cases} x_k^+ = x_k + \delta x_k \text{ right rapid pounce} \\ x_k^- = x_k - \delta x_k \text{ left rapid pounce} \\ x_k \text{ no pounce} \end{cases} \qquad (4)$$

The sequence of parameters $\delta x_k$ depending on the fitness $g_k$ is introduced in Figure 2. Each agent of the $\tilde{G}_1$-team evaluates its next move based on the mechanism of Equation (4) in order to increase the FOM. This is done iteratively, starting from the leader (agent with the highest value in the $g_k$-sequence) and going down the ranks of the pack fitness hierarchy, until all of the wolves locations in the group are updated. It is worth noting that for any agent of this group, at a given iteration $t$, the choice of the next best value of each variable $x_k$ required three evaluations of the FOM: $x_k = best\{x_k^- \text{ left}, x_k^+ \text{ right}, x_k \text{ no}\}$ pounce. This routine can be completely parallelized.

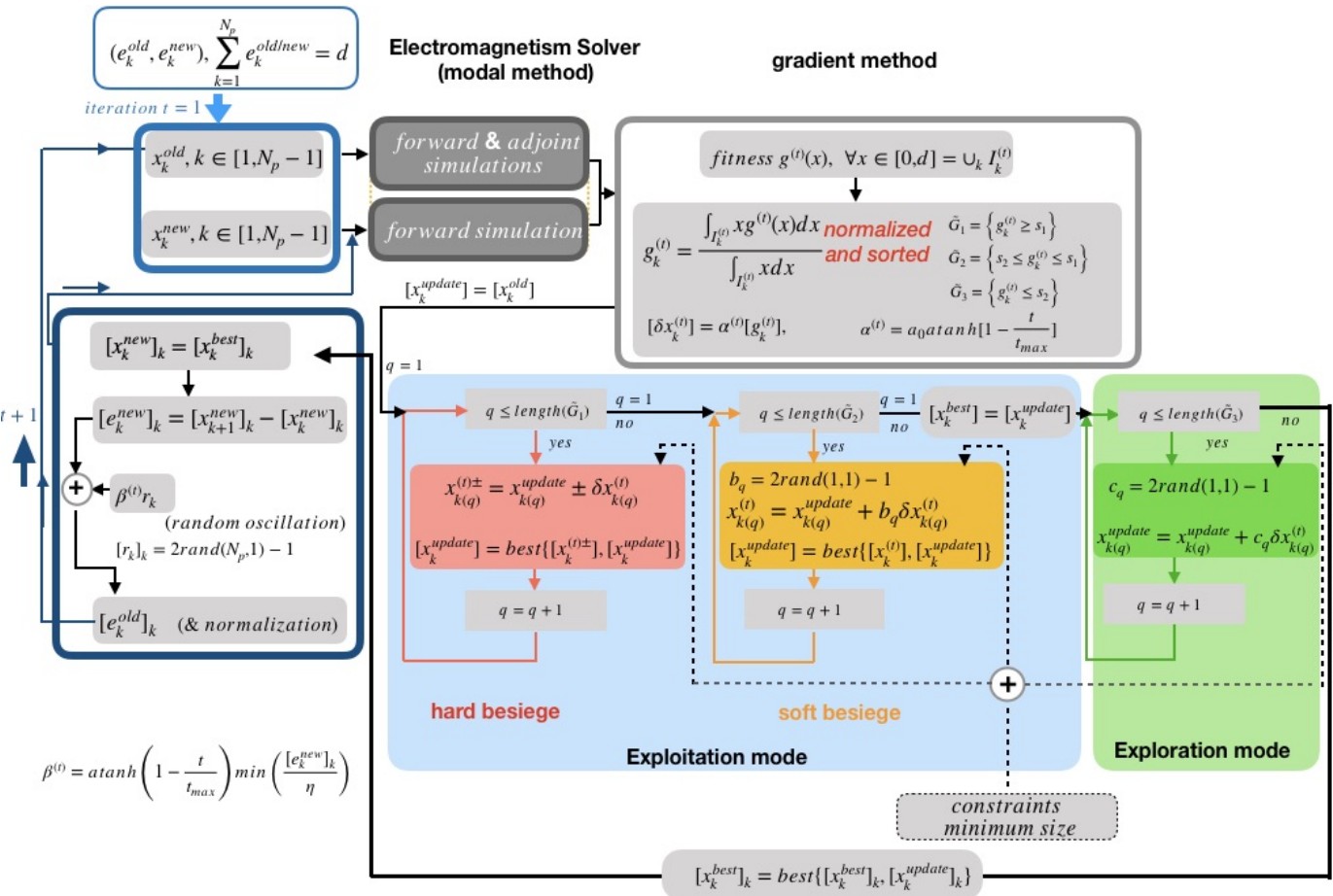

**Figure 2.** Flowchart of the proposed optimization method using a wolf pack biomimicry-gradient-based strategy.

### 2.3.2. Team $\tilde{G}_2$ Strategy: Exploitation Mode with a Soft Besiege

Agents of this team are also actively pursuing the prey but being slightly behind the members of the leading $\tilde{G}_1$-team. Therefore the prey should be less sensitive to any variation in their current locations. While pursuing their prey and trying to encircle it, these agents can allow themselves some slight random left or right rapid pounces around their current positions, and compare to their previous locations (no pounce), while improving

the FOM. In this case, the next position of each wolf is updated using the following Equation (5):

$$x_k^{update} = \begin{cases} x_k + b_k \delta x_k, b_k \in [-1,1] \text{ random left or right rapid pounce} \\ \\ x_k \text{ no pounce} \end{cases} \tag{5}$$

where $b_k = 2r_k - 1$, $r_k \in [0,1]$ being a random number. Let us highlight that this second team also operates in exploitation mode. However, contrary to the $\tilde{G}_1$ team, for any agent, only two evaluations of the FOM (parallel computing) are performed, in order to estimate the next best position.

### 2.3.3. Team $\tilde{G}_3$ Strategy: Exploration Mode

Finally, the $\tilde{G}_3$ team is made up of agents whose movements have very little impact on the process in progress. Therefore, these agents can allow a collective random positioning before evaluating at once the effectiveness of this global action. If this collective action does not effectively modify the figure of merit then the whole group collectively returns to its initial position. This strategy can mathematically formulate as follows:

$$[x_k^{update}]_k = \begin{cases} [x_k]_k + [c_k][\delta x_k]_k \text{ random rapid pounces for the whole sequence} \\ [x_k]_k \text{ no pounce} \end{cases} \tag{6}$$

where $c_k$ is a sequence of random numbers in $[-1,1]$.

At the end of all these three-teams actions, a new geographic distribution $[e_k^{new}]_k$ can be updated as

$$e_k^{new} = x_k^{update} - x_{k-1}^{update}, k \in [1, N_{p+1}] \tag{7}$$

### 2.4. Constraints Inside the Pack

During hunting, any wolf must update at any time information about all hunter agents, but also, while moving towards the prey, a minimum distance to the closest neighbors must be kept. This minimum distance between the wolves of the pack is required to avoid collisions between neighboring agents, but above all, it allows guaranteeing a better local range of action for all the members of the pack. A maximum distance constraint can be also enforced in order to preserve the overall cohesion of the group and prevent the prey from escaping between the lines.

### 2.5. Repeated Escape Attempt of the Prey

When the wolf pack attacks, the prey naturally attempts to escape from this threatening situation. Different situations occur, according to the current state of the chasing, but one of the most dangerous situations for the prey is that of encirclement. To escape from this situation, the prey performs some rapid moves, the intensity and success of which strongly and mainly depend on how much energy it has left. This prey's residual energy, denoted $\beta^{(t)}$, decays with respect to the iteration $t$. $\beta^{(t)}$ may also depend on several other factors such as the maximum number of iterations $t_{max}$, the density of the wolf pack characterized by the minimum distance between two wolves. This is modeled as:

$$\beta^{(t)} = atanh\left(1 - \frac{t}{t_{max}}\right) min\left(\frac{[e_k^{new}]_k}{\eta}\right) \tag{8}$$

The term $min\left(\frac{[e_k^{new}]_k}{\eta}\right)$ holds information about the minimum distance between two wolves. The parameter $\eta$ drives the density of the wolf pack. This repeat escaping action of

the prey leads to a disturbance of the old wolf pack features. These geographic features are then re-evaluated through the following formulas:

$$e_k^{old} = e_k^{new} + \beta^{(t)}(2r_k - 1);$$ (9)

where $r_k \in [0, 1]$ is a random number that changes at any iteration $t$. The flowchart of the proposed geometric parameters optimization method based on wolf pack biomimicry hunt strategy is presented in Figure 2. In the next section, the proposed algorithm is applied to design plasmonic lenses, consisted of a relatively high number of design elements (parameters), enabling near and far field focusing.

## 3. Results and Discussions

### 3.1. Numerical Method: Polynomial Modal Method

Let us consider in a Cartesian coordinates system $(\mathbf{e_x}, \mathbf{e_y}, \mathbf{e_z})$ the structure of Figure 3a which is consisted of a metallic film with arrayed nanoslits (air-gap) with a constant depth $h$ but variant widths $(e_k)_k$. Our design's objective consists of optimizing the $(e_k)_k$-sequence in order to obtain a focusing of the incident field at a desired space location.

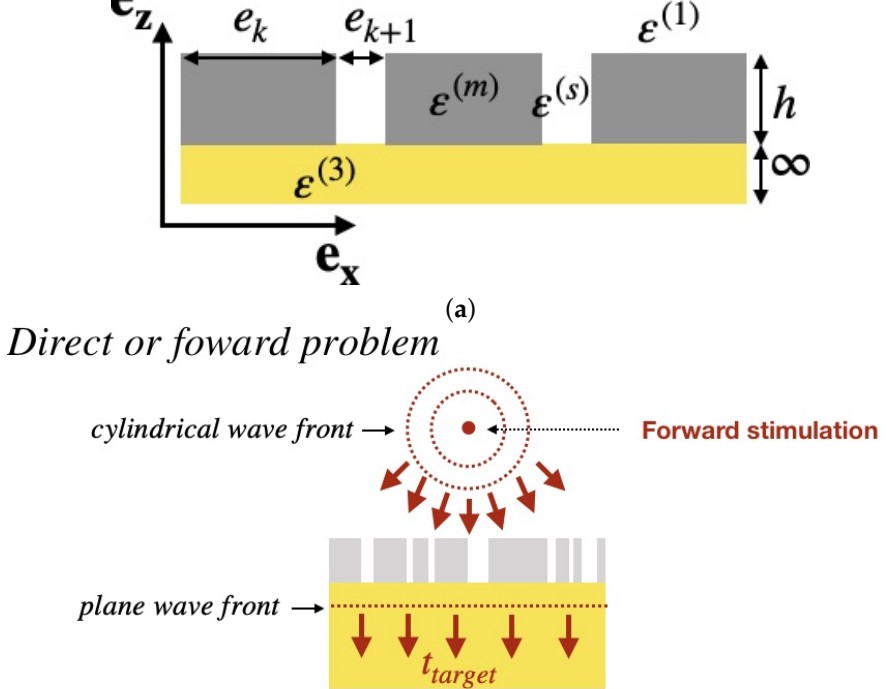

(a)

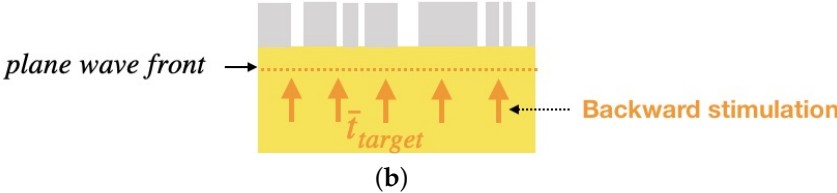

(b)

**Figure 3.** design of a one dimensional flat metalens. (**a**) Schematic of dispersive metal film perforated with a non periodic subwavelength array of 1D nanoslits. (**b**) the adjoint-based optimization method involved two different sources: an electric dipole and a polarized incident plane wave. (**a**) Sketch of the plasmonic dispersive metalens; (**b**) Assessment of the FOM variation computing.

The Aperiodic Polynomial Modal Method (APMM) is used to perform the forward and the backward problems in two simulations. See Figure 3b. For audiences who are not familiar with this method, let us briefly outline the main steps of the polynomial Modal Method (PMM) based on Gegenbauer polynomials $G_n^{\Lambda}(x)$ expansion [22–26]. In the PMM/APMM, the structure is divided into $N_p$ intervals $I_x^{(k)}, k = 1 : N_p$ in the $(Ox)$ direction and $N_z$ layers $I_z^{(l)}, l = 1 : N_z$ in $(O, z)$ direction. In the current example $N_z = 3$. In each layer $I_z^{(l)}$, each component of the electromagnetic field is expanded on a set of eigenfunctions, i.e.; solutions $|\psi_q\rangle$ of an eigenvalue equation; that is in the current case, the following TM propagation equation Equation (10):

$$\mathcal{L}^{(l)}(\omega)|\psi_q^{(l)}(\omega)\rangle = (\gamma_q^{(l)}(\omega))^2|\psi_q^{(l)}(\omega)\rangle \tag{10}$$

with

$$\mathcal{L}^{(l)}(x,\omega) = \left(\frac{c}{\omega}\right)^2 \varepsilon^{(l)}(x,\omega)\partial_x \frac{1}{\varepsilon^{(l)}(x,\omega)}\partial_x + \varepsilon^{(l)}(x,\omega). \tag{11}$$

In each layer $l$ the $H_y^{(l)}(x,z)$ component is then written as linear combination of the eigenfunctions $\psi_q$ of Equation (10):

$$H_y^{(l)}(x,z,\omega) = \sum_q A_q^{(l)}(\omega)e^{-ik_0(\omega)\gamma_q^{(l)}(\omega)z}\psi_q^{(l)}(x,\omega). \tag{12}$$

The relative permittivity of the upper and the lower regions are denoted by $\varepsilon^{(1)} = 1$ and $\varepsilon^{(3)} = 1.45$ respectively. Silver and gold are considered in our simulations and the Drude-Lorentz model [27,28] is used to describe their dispersive relative permittivity function $\varepsilon^{(metal)}$. Our design objective involves two electromagnetic sources: an electric dipole source for the direct or forward problem and a polarized incident plane wave for the reciprocal or backward problem. The forward and backward configurations, used to compute the variation of the FOM are sketched in Figure 3b. Practical details for the numerical treatment of the dipole source implementation in modal method can be found in [29].

### 3.2. Plasmonic Lens Design

We design metalenses consisting of a perforated metal film, capable of focusing a normally incident TM plane wave (with $\lambda = 0.637$ μm), at a focal distance $z_f = 15\lambda$ in the vacuum ($\nu^{(1)} = 1$). The perforated film with a height $h = 400$ nm is deposed on a SiO$_2$ substrat. To ensure the convergence of the polynomial expansion, $n = 4$ polynomials are used on each subinterval $I_x^{(k)}, k = 1 : N_p$.

As initial conditions of the algorithm, we use 10 various random initial sequences of nsnr (nanoslits and nanoridges) with elements widths $[(e_k^{old}, e_k^{new})]_k$, generated according to equation Equation (1). The maximal number of iterations $t_{max}$ is set to 20. In this first example, each initial layout is made of $N_p = 105$ nsnr elements distributed on a d = 10.75 μm-wide interval. The minimum size constraint is set to 50 nm. The other numerical used-defined parameters are $\alpha_0 = 0.1$, $s_1 = 25\%$ and $s_2 = 1\%$. Results are presented in Figure 4. These figures show, in the focal planes, the line scans of the normalized magnetic field intensity along the $Ox$-axis ($z = 0$) (Figure 4a) and along the $Oz$-direction $x = 0$ (Figure 4b). Although any gradient-based method is known to be very sensitive to the initial conditions, as show these figures, for all the ten random initial conditions, that a very large part of the ten designed metalenses maintain satisfactory efficiencies close to 30% for the current lossy plasmonic lens. Both the lens size and the number $N_p$ of nsnr elements can be exploited to manage the focus spot size as is clearly shown in Figure 5. In these figures, we plot, the line scans of the normalized magnetic field intensity in the transverse ($z_f = 0$) (Figure 5a) and in the longitudinal $x = x_0$ (Figure 5) focal plane respectively, for four values of the couple $(N_p, d)$:

$(N_p, d) \in \{(105, 10.75\ \mu m), (125, 12.75\ \mu m), (165, 16.75\ \mu m), (185, 18.75\ \mu m)\}$. Regarding these results, the ratio of the focal intensity to the intensity of incident wave increases with an increasing number of $(N_p, d)$ couples. In the case of $(N_p, d) = (105, 10.75\ \mu m)$, this efficiency is close to 29%. While it reaches 38% for a 18.75 μm-width lens composed of $N_p = 185$ metallic-nanorods/nanoslits.

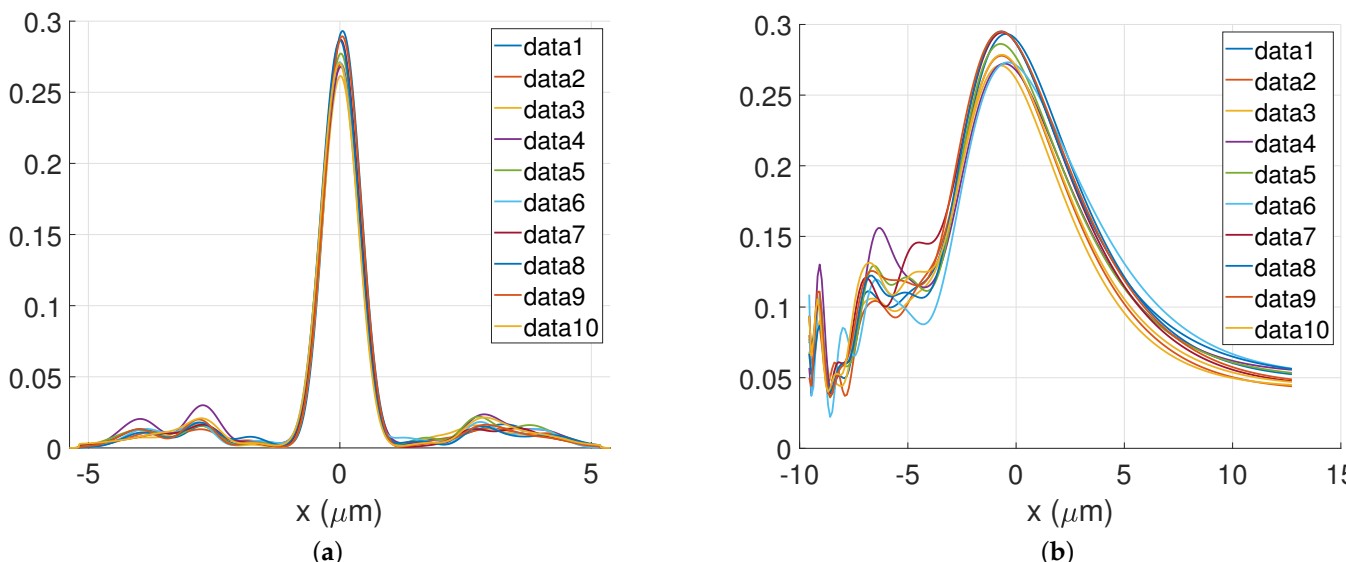

**Figure 4.** Design of a 10.75 μm-wide plasmonic metalens consisting of $N_p = 105$ silver-nanoridges and nanoslits focusing a normal incident TM-polarized plane wave at a wavelength $\lambda = 0.637$ μm at a distance of $15\lambda$. Line scans of the normalized magnetic field intensity in the transverse ($z_f = 0$) (**a**) and in the longitudinal $x = 0$ (**b**) focal plane respectively, for ten random realizations of initial conditions. Numerical parameters: $\lambda = 0.637$ μm, $N_p = 105$, $h = 400$ nm, TM polarization. (**a**) $|H_y(x, 0)|^2 / \int_{\mathcal{I}} |H_y^{inc}(x, z_s)|^2 dx$; (**b**) $|H_y(x_0, z)|^2 / \int_{\mathcal{I}} |H_y^{inc}(x, z_s)|^2 dx$.

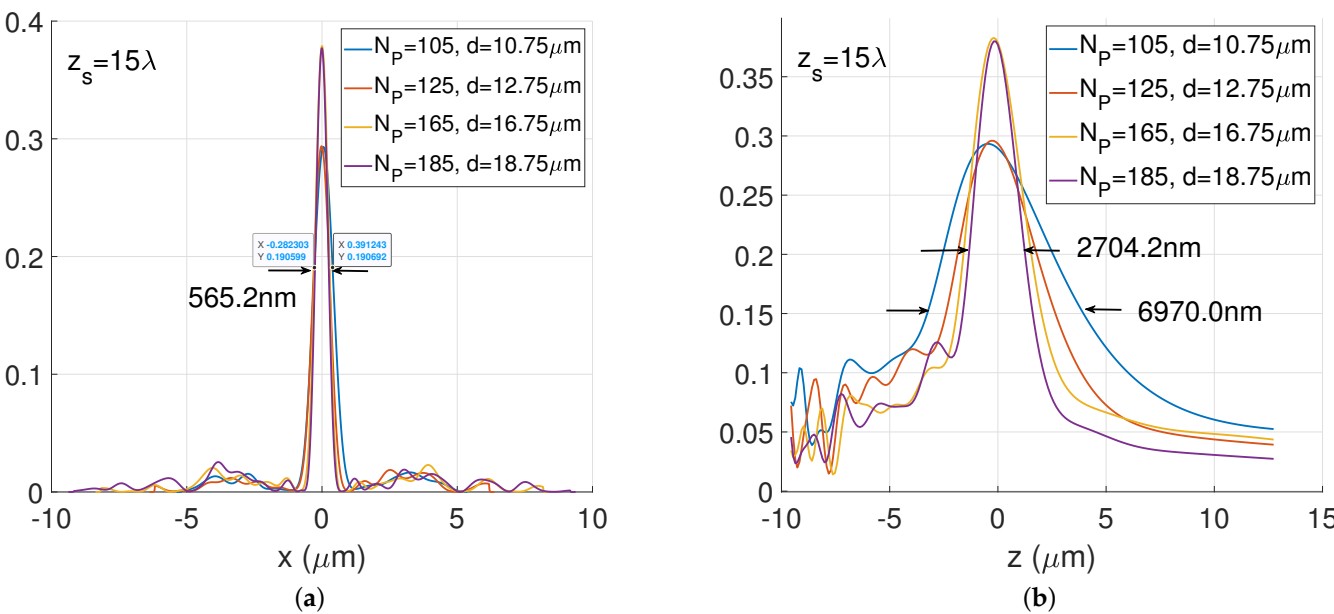

**Figure 5.** Design of $d$-wide silver plasmonic metalense consisting of $N_p$ silver-nanoridges and air-gap focusing a normal incident TM-polarized plane wave at a wavelength $\lambda = 0.637$ μm at a distance of $15\lambda$. Line scans of the normalized magnetic field intensity in the transverse ($z_f = 0$) (**a**) and in the longitudinal $x = x_0$ (**b**) focal plane respectively, for four values of the couple $(N_p, d)$: $(N_p, d) \in \{(105, 10.75\ \mu m), (125, 12.75\ \mu m), (165, 16.75\ \mu m), (185, 18.75\ \mu m)\}$. Numerical parameters: $\lambda = 0.637$ μm, $h = 400$ nm, TM polarization. (**a**) $|H_y(x, 0)|^2 / \int_{\mathcal{I}} |H_y^{inc}(x, z_s)|^2 dx$; (**b**) $|H_y(x_0, z)|^2 / \int_{\mathcal{I}} |H_y^{inc}(x, z_s)|^2 dx$.

Now we analyse the impact of the $(N_p, d)$ couple on both the LFWHM (lateral full width half maximum) and on the AFWHM (axial full width half maximum). As shown in Figure 5a,b, the LFWHM hardly depends on the lateral feature of the metalens. In Figure 5a the LFWHM is approximately equal to 565.2 nm. Contrary to the LFWHM, the AFWHM strongly depends on the width of the metalens. As shown in Figure 5, the AFWHM decreases when the lens size $d$ and number of patterns $N_p$ increase. For example, in the case of the silver-plasmonic metalens considered in our investigation, the AFHW is equal to 6916.9 nm for $(N_p, d) = (105, 10.75 \ \mu m)$ and it decreases to 2704.2 nm for $(N_p, d) = (185, 18.75 \ \mu m)$, yielding a better high-quality cylindrical focal spot.

It is worth noting that some optimized devices yield original unexpected asymmetric on-axis metalenses, even though they are under a normal incidence. Figure 6a,b presents the cartography of the normalized magnetic field's intensity in the plane $(X, O, Z)$ associated with the optimized geometry of Figure 6a. In Figure 6a, we report the values of the (nanorodes, nanoslits)-widths sequence $[e_k]_k$ with respect to their locations labelled by $k$. No symmetry property can be extracted from these results, indicating that sophisticated energy exchange between the modes in the design area takes place and these interactions cannot be explained or/and modeled thanks to some trivial rules. The modes contained in the design area of the structure are not symmetric but their combination yields a symmetric spatial distribution of the electromagnetic far field. Without enforcing some symmetry properties, the optimization process freely evolves towards unexpected exotic structures. From the above study, we anticipate that a high-quality focusing spot can be reached by increasing the size and the nsnr-number $N_p$ of the structure under optimization. However, numerically speaking, increasing the lens's geometric features is computationally expensive, since both the number of the design variables $N_p$ and the number of basis functions required for the electromagnetic field numerical approximation increase also. To circumvent this drawback, we introduce a computationally less expensive approach termed the two-meta-cells-stitched approach. In this approach, a metalens constituted of $N_p$-nsnr elements is viewed as an arrangement of two interlocked subsections, $[A, D] \cup [B', A']$ denoted meta-cells, as is shown in Figure 7. Only the meta-cell $[A, D]$ constituted of $n_p$ nsnr-elements is simulated as a off-axis aperiodic metalens. The complementary $n_p - 2$ meta-cell $[B', A']$ is built as the symmetric of the a $n_p - 2$ nsnr elements of a sub-meta-cell $[A, B]$ of the meta-cell $[A, D]$ ($[A, B] \cup [A, D]$).

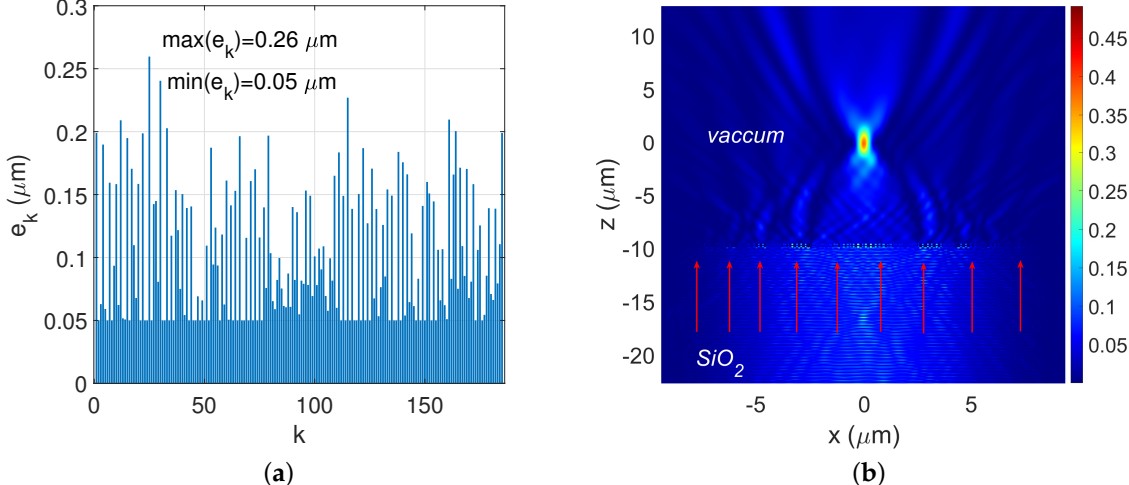

(a)  (b)

**Figure 6.** Design of a plasmonic silver-flat-metalens of $d = 18.75 \ \mu m$-wide consisting of $N_p = 185$ silver-nanoridges and air-gap focusing a normal incident TM-polarized plane wave at a wavelength $\lambda = 0.637 \ \mu m$ at a distance of $15\lambda$. (**a**) shows the values of the (nanorodes, nanoslits)-widths sequence $[e_k]_k$ with respect to their locations labelled by $k$. (**b**) presents the cartography of the normalized magnetic field's intensity in the plane $(X, O, Z)$. Numerical parameters: $\lambda = 0.637 \ \mu m$, $N_p = 52$, $e_z = 400$ nm, TM polarization. (**a**) $|H_y(x_0, z)|^2 / \int_{\mathcal{I}} |H_y^{inc}(x, z_s)|^2 dx$; (**b**) $|H_y(x, 0)|^2 / \int_{\mathcal{I}} |H_y^{inc}(x, z_s)|^2 dx$.

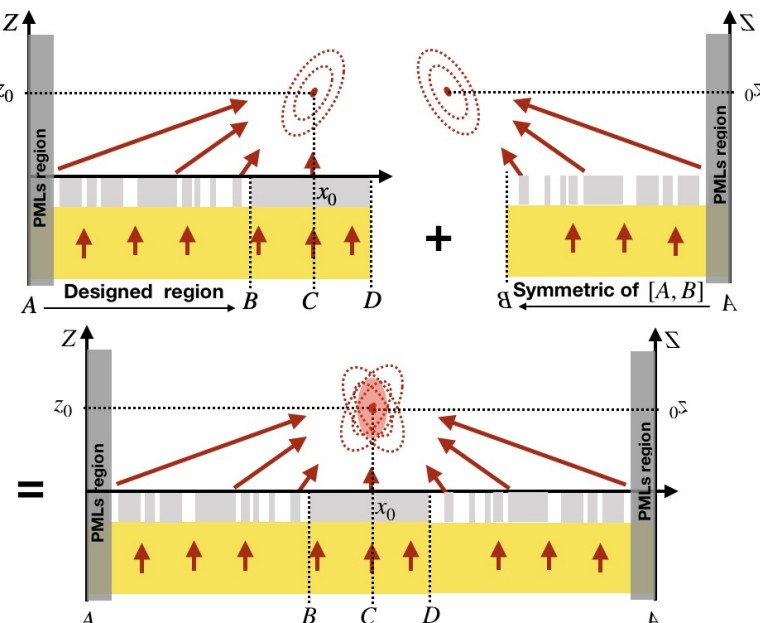

**Figure 7.** Design of a one dimensional flat metalens: illustration of the two-metacells-stitched approach.

To illustrate the efficiency of the above idea of two-meta-cells stitched principle, we design two types of metalenses of different widths, $d = 24.99$ μm and $d = 41$ μm constituted of $N_p = 248 = 125 + 125 - 2$ and $N_p = 408 = 205 + 205 - 2$ nsnr elements respectively. The other parameters are $h = 400$ nm, and the focal length is set to $15\lambda$. For each example, first we design a $L$-wide $[A, D]$ metacell to achieve efficient far field focusing of the transmitted field at $(x_0, z_s)$ so that the focus spot is located at the extreme limit of the computation domain. Typically, $x_0$ is chosen very close to $L$. Second, the second sub-metacell namely $[B', A']$ is created from the previous optimized off-axis metalens $[A, D] = [A, B] \bigcup [B, D]$ as the symmetric of the $[A, B]$ subsection. The completed final on-axis device is then obtained by stitching together the $[A, D]$ and $[B', A']$ subsections. First, Let us consider the case of 24.99 μm-wide metalens made of 248 nsnr elements which $[A, D]$ subsection is a 12.55 μm-wide metacell made of 125 nsnr elements. Figure 8a presents the normalized transmitted field's intensity distribution on $(X, O, Z)$ plane. A clear off-axis focus spot appears at the predicted location, which agrees with our design's objective. Let us also notice a transmission efficiency of about 25%. Now let us apply the stitching operation as described previously. The result of this post-processing is displayed in Figure 8b where we plot the field's intensity distribution of the completed lens excited by an incident TM plane wave at normal incident. It is worth noting that this two metacells stitched process leads to an increase, in both the size and the number of the constituent elements of the metalens. In this first example, a $(L, n_p) = (12.55$ μm, $125)$ metacell yields $(d, N_p) = (24.99$ μm, $248)$ metalens with more than two times transmission efficiency enhancement: the efficiency of the stitched device increases from 25% to 60%. Figure 8c,d show the values of the (nanoridges, nanoslits)-widths sequence $[e_k]_k$ with respect to their locations labelled by $k$. Now let us increase the metalens feature, i.e., both the size and the number of nsnr elements. In this second example, the structure is a $d = 41$ μm-wide metalens constituted of $N_p = 408 = 205 + 205 - 2$ nsnr elements. The primary metacell, i.e., the $[A, D]$ subsection, is specifically designed as a 20.55 μm-wide off-axis metalens with 205 nsnr elements. Figure 9a presents the normalized transmitted field's distribution. The deflection of focal point to the extreme limit of the design domain, with an efficiency of 30% is clearly visible. As shows Figure 9b, by stitching together both aperiodic metacells, namely the subsection $[A, D]$ and $[B', A']$, the on-axis intensity is highly improved from 30% to 70%. Figure 9c,d show the values of the (nanoridges, nanoslits)-widths sequence $[e_k]_k$ with respect to their locations labelled by $k$.

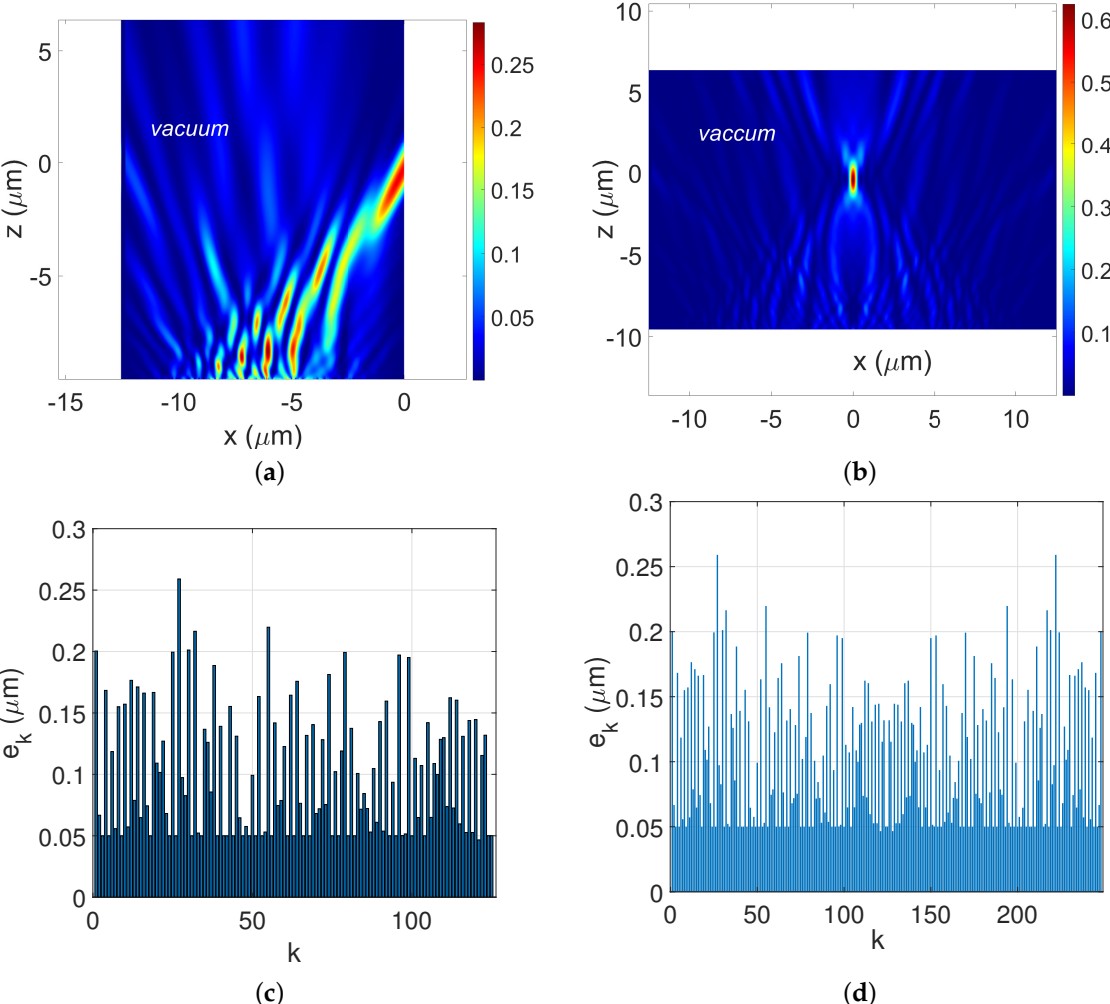

**Figure 8.** Design of a plasmonic silver-flat-metalens of $d = 24.99$ μm-wide consisting of $N_p = 248$ silver-nanoridges and nanoslits focusing a normal incident TM-polarized plane wave at a wavelength $\lambda = 0.637$ μm at a distance of $15\lambda$. (**a**,**b**) presents the cartography of the normalized magnetic field intensity in the plane $(X, O, Z)$. (**c**,**d**) shows the values of the (nanoridges, nanoslits)-widths sequence $[e_k]_k$ with respect to their locations labelled by $k$. Numerical parameters: $\lambda = 0.637$ μm, $N_p = 125$, $e_z = 400$ nm, TM polarization. (**a**) $d = 12.55$ μm; (**b**) $d = 24.99$ μm; (**c**) $n_p = 125$; (**d**) $N_p = 248 = 125 + (125 - 2)$.

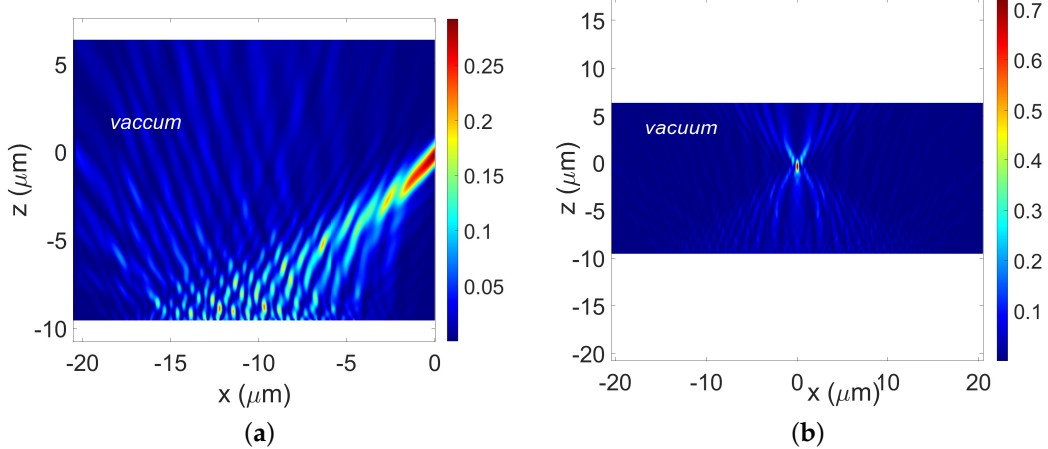

**Figure 9.** *Cont.*

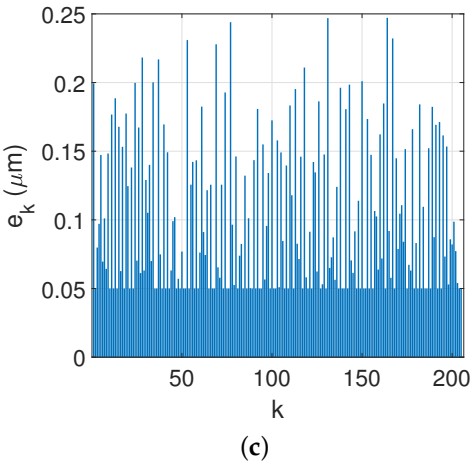
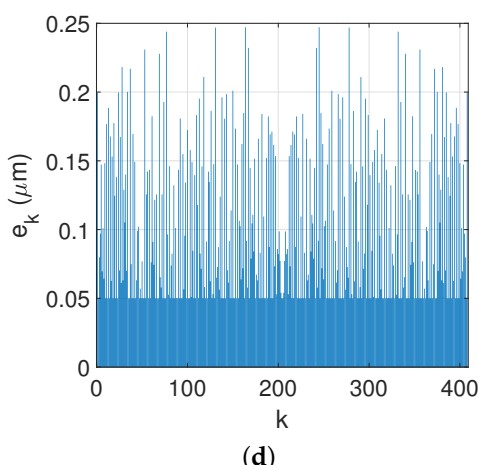

(c)　　　　　　　　　　　　　　　　　　　　(d)

**Figure 9.** Design of a plasmonic silver-flat-metalens of $d = 41$ μm-wide consisting of $N_p = 248$ silver-nanoridges and nanoslits focusing a normal incident TM-polarized plane wave at a wavelength $\lambda = 0.637$ μm at a distance of $15\lambda$. (**a**,**b**) presents the cartography of the normalized magnetic field intensity in the plane $(X, O, Z)$. (**c**,**d**) shows the values of the (nanoridges, nanoslits)-widths sequence $[e_k]_k$ with respect to their locations labelled by $k$. Numerical parameters: $\lambda = 0.637$ μm, $N_p = 205$, $e_z = 400$ nm, TM polarization. (**a**) $d = 20.55$ μm; (**b**) $d = 41$ μm; (**c**) $n_p = 205$; (**d**) $N_p = 408 = 205 + (205 - 2)$.

Finally we present a case of an optical near-field subwavelength spatial focusing originated from the interference of higher spatial harmonic field distribution. The focus length is set $z_s = 3\lambda$ in the free space. Since the foci is closed to the structure, it is possible to obtain a high-focusing of an incident light by enhancing the near field (evanescent field) contribution. We show in Figure 10 the results of a configuration of a gold metalens. As can be seen in Figure 10a,b, a subwavelength LFWHM can be reached: the LFWHM is equal to 304.5 nm $\leq \lambda/2$ in the case of the gold plasmonics metalens designed here. Figure 11a presents the cartography of the normalized magnetic field intensity in the plane $(X, O, Z)$. Figure 11b shows the values of the (nanoridges, nanoslits)-widths sequence $[e_k]_k$ with respect to their locations labelled by $k$. Some other results not presented in this paper, including information of required cputimes, some .gif animations, are available on https://photonicsnum.com (accessed on 15 May 2021).

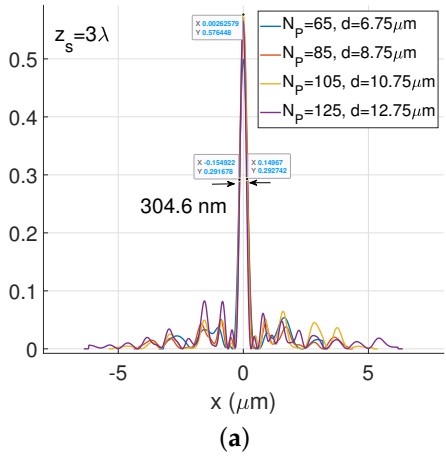
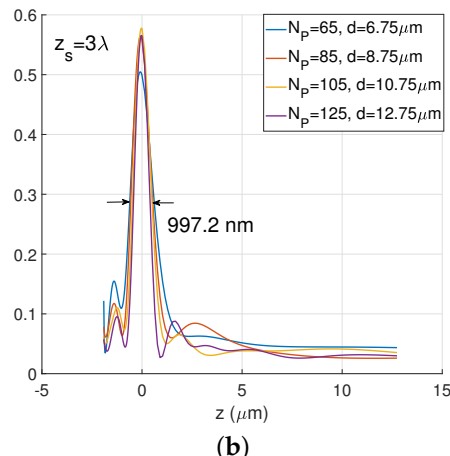

(a)　　　　　　　　　　　　　　　　　　　　(b)

**Figure 10.** Design of $d$-wide gold plasmonic metalens consisting of $N_p$ gold nanoridges and nanoslits focusing a normal incident TM-polarized plane wave at a wavelength $\lambda = 0.637$ μm at a distance of $3\lambda$. Line scans of the normalized magnetic field intensity in the transverse ($z_f = 0$) (**a**) and in the longitudinal $x = x_0$ (**b**) focal plane respectively, for four values of the couple $(N_p, d)$: $(N_p, d) \in \{(65, 6.75 \text{ μm}), (85, 8.75 \text{ μm}), (105, 10.75 \text{ μm}), (125, 12.75 \text{ μm})\}$. Numerical parameters: $\lambda = 0.637$ μm, $h = 400$ nm, TM polarization. (**a**) $|H_y(x,0)|^2 / \int_{\mathcal{I}} |H_y^{inc}(x, z_s)|^2 dx$; (**b**) $|H_y(x_0, z)|^2 / \int_{\mathcal{I}} |H_y^{inc}(x, z_s)|^2 dx$.

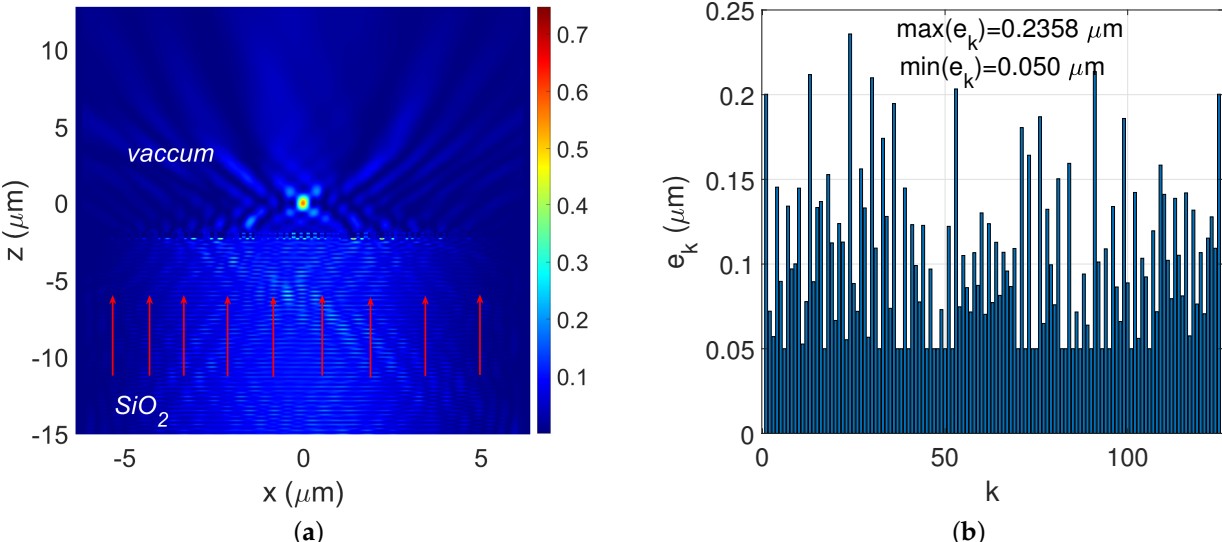

**Figure 11.** Design of a plasmonic gold-flat-metalens of $d = 12.75$ μm-wide consisting of $N_p = 125$ gold-nanoridges and air-gap focusing a normal incident TM-polarized plane wave at a wavelength $\lambda = 0.637$ μm at a distance of $3\lambda$. (**a**) presents the cartography of the normalized magnetic field intensity in the plane $(X, O, Z)$. (**b**) shows the values of the (ridges, air-gaps)-widths sequence $[e_k]_k$ with respect to their locations labelled by $k$. Numerical parameters: $\lambda = 0.637$ μm, $N_p = 125$, $e_z = 400$ nm, TM polarization. (**a**) $|H_y(x, 0)|^2 / \int_{\mathcal{I}} |H_y^{inc}(x, z_s)|^2 dx$; (**b**) $|H_y(x_0, z)|^2 / \int_{\mathcal{I}} |H_y^{inc}(x, z_s)|^2 dx$.

## 4. Conclusions

In this paper, we reported a biomimicry-gradient-based optimization algorithm. The method was successfully applied to perform the design of a functional metasurface which is a plasmonic metalens enabling a high energy flow focusing on a well-defined cylindrical focus spot. This is accomplished through the introduction of a set of a few, but key, design parameters allowing the fine-tuning of a given metasurface geometrical features. These parameters are updated exploiting both available gradient information and intelligent cooperative chase strategy. The full plasmonic device is optimized in one time, and showed that the efficiency of the optimized device is less sensitive to initial conditions. We also show that the proposed approach provides both expected symmetric devices and more creative unexpected asymmetric on-axis metalenses; even though under normal illumination.

There is no unique and universal method that allows efficiently solving PDEs obtained from Mawxell's equations in the general case. However, for certain categories of electromagnetic problems with specific geometries and symmetries, it is possible to identify a class of numerical methods allowing solving accurately and efficiently these PDEs. One-dimensional metasurfaces often consist of an arrangement of rods deposited on a substrate, and Modal Methods have proven to be very effective in solving such problems. These approaches consist in describing the electromagnetic field in terms of the eigenfunctions of an operator that represents the propagation through the structure. These eigenfunctions are expanded on a set of a given basis of functions. For example in the particular case of the Fourier Modal Method (FMM), the Fourier basis functions are used. However, the main drawback of the FMM [30–33] remains the representation of non smooth fields through finite Fourier sums resulting, inevitably, in slow convergence rates. In addition, the worst cases are those of diffraction problems involving metals structures. These cases often require a high number of Fourier basis functions to converge which increases the computational times. To remedy this situation, we replace, the FMM with the Polynomial Modal Method equipped with Perfectly Matched Layers (PMLs). The proposed method, which is expected to contribute to the effort of photonics novel devices inverse design novel, proves to be suitable and robust in designing micron-scale devices. However, it cannot be extended at it is to the design of very large-scale structure without some approximations. In regards of positive outlook, we envision extending the method to design such

structures namely 2D and very large-scale structures, in the framework of modal method, by incorporating the stitching technique, presented in [8]. In addition, we expect this will help to quickly and efficiently solve some classes of current photonics concern.

**Funding:** This work has been sponsored by the French government research program "Investissements d'Avenir" through the IDEX-ISITE initiative 16-IDEX-0001 (CAP 20-25).

**Institutional Review Board Statement:** Not applicable.

**Informed Consent Statement:** Not applicable.

**Data Availability Statement:** Not applicable.

**Conflicts of Interest:** Author declares there is no conflicts of interest.

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
