# Peer review of "Biomimicry-Gradient-Based Algorithm as Applied to Photonic Devices Design: Inverse Design of Flat Plasmonic Metalenses"

_applsci, doi:10.3390/app11125436_

Round 1

Reviewer 1 Report

The paper is interesting and well written. The method for the design of photonic devices is described in detail and applied for the design of a focusing device. I recommend publication after the following minor issues will be addressed:

1-In Section 3.2 is described a plasmonic lens design. However there is no information about the type of metal nor on the surrounding materials. The authors should include these information and a discussion of how the results might depend on the environment.

2- Why in all the figures it is shown the magnetic field intensity instead of the electric field?

3-Figure 7 must be improved. The right half of the figure is reverse and it is hard to read the content. Furthermore in the text it is called A'B' while in the figure is named BA (with reversed letters).

4- The author should include a comparison with other existing competing methods for the design of photonic devices, highlighting pros and cons of the proposed method.

5- No information at all is provided on the software used to run the algorithm, if it is developed by the author (and if so the author should specify which programming environment has been used) or if it is based on other software (in this case the author should provide appropriate references). Also, the software used to evaluate the electromagnetic field of the obtained design is not specified. The author should include all the relevant information about the different software used for the simulations.

6- Please include information also about the memory usage and the computational time of the algorithm to get the illustrated designs.

7- I would not use the name "metasurface" or "metalens", as it is done somewhere in the paper, since it is technically inappropriate.

Author Response

Dear reviewer

First of all, I would like to thank you for your constructive remarks and questions that help to improve the quality of the article. In the following I respond to the points you raised  and explain (when it is the case) the changes made in the paper. These changes are displayed in blue color in the revised version of the paper. Please see my respond in the cover letter below. Pages 1:9 are devoted to my response, followed by the revisited manuscript at page 10.

Reviewer 2 Report

The manuscript is devoted to biomimicry-gradient-based algorithm as applied to photonic devices design. The author used the method to design a non-periodic metasurface consisting of plasmonic metalenses. The concept is interesting and the article has been well thought out and carefully prepared. Numerical results also proved the proposed algorithm. Despite this, I believe its current form requires a few corrections. Therefore, I am asking the author to refer to the following points:
1. There is a typo in the 6th line of text: „senarios” (should be: scenarios).
2. The author wrote that he applied the algorithm to “a non-periodic metasurface”. From my perspective, this term is very debatable. Note that the definition of metamaterials says that they are periodic structures and the period must be several times smaller than the incident wavelength. Can metamaterial be non-periodic? This point should be carefully discussed.
3. What is the complexity of the computational algorithm of the proposed method compared to other most popular and frequently used methods (i.e. FDTD, TMM, etc.). Please see: Appl. Sci. 2017, 7, 618; Liq. Cryst. 42 (4), 430-434, (2015); IEEE Access, vol. 2, pp. 437- 450, 2014, etc. I miss here such a reference and comparison to the most frequently used methods for simulating photonic meta-devices. Of course, the author lists many advantages of the proposed concept. Nevertheless, I would like the reader, who uses other methods, is able to clearly define the advantages and disadvantages of particular algorithms. Please discuss it in detail.
4. Please adapt the legend in Fig. 4. to the content of the article. In this form, it can be misleading.
5. The captions under Figs. 8. and 9. are incorrect. Please correct them.
6. In conclusion, it is worth mentioning the limitations of the described algorithm. What are its main disadvantages?

Author Response

First of all, I would like to thank you for your constructive remarks and questions that help to improve the quality of the article. In the following I respond to the points you raised  and explain (when it is the case) the changes made in the paper. These changes are displayed in blue color in the revised version of the paper. Please see my respond in the cover letter below. Pages 1:9 are devoted to my response, followed by the revisited manuscript at page 10.
